# ALPHASURF: ON-THE-FLY SURFACE COMPUTATIONS FOR PROTEIN REPRESENTATION LEARNING

**Victor Gertner[1,2,3], Frédéric Cazals[4,†], Vincent Mallet[1,2,3,†]**

[1]Mines Paris, PSL Research University, CBIO, Paris, France;
[2]Institut Curie, PSL Research University, Paris, France;
[3]INSERM, U1331, Paris, France;
[4]Centre Inria at Université Côte d'Azur, France;

[†]Equal supervision.

vincent.mallet@minesparis.psl.eu

## ABSTRACT

Several protein surfaces have been proposed for visualization purposes, and more. Recently, machine learning approaches incorporating surfaces as a biomolecular representation have emerged with strong performances, at the cost of increased computations. We show that this burden can be avoided, computing on-the-fly a 2D simplicial complex, with no overhead and no performance loss.

## 1 MOTIVATION AND ORGANIZATION OF THIS PAPER

Various protein surfaces have been developed in the last four decades, following their initial introduction in Connolly (1983). They represent a foundational tool for visualization (DeLano et al., 2002) and geometry processing (Gerstein & Richards, 2001; Loriot & Cazals, 2010). More recently, surfaces computed with MSMS (Sanner et al., 1995) were split into patches and successfully used within a learning pipeline (Gainza et al., 2020). Further learning applications used whole surfaces, leveraging geometric deep learning advances, and resulting in state-of-the-art performance (Mallet et al., 2024).

The use of surfaces derived from MSMS has a high compute and memory footprint. Therefore, cheaper proxies were proposed, such as points lying near the surface and their approximate normal, albeit with limited performance (Sverrisson et al., 2021). **We argue this limitation has no foundational basis** and stems only from the large adoption of MSMS as the default tool to build surfaces.

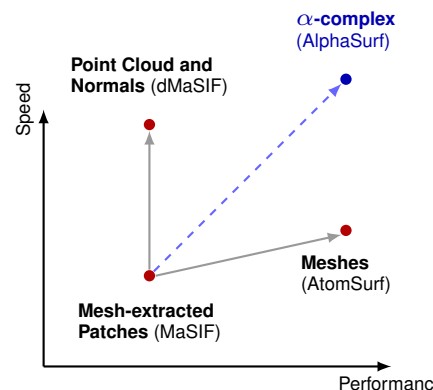

Figure 1: Different molecular surfaces trade off speed and performance. AlphaSurf gets the best of both worlds.

In this paper, we start by formally introducing the mathematical tools used in surface computations (Section 2). We proceed by rigorously reviewing the different surface constructions and empirically evaluate their computational aspects (Section 3). Finally, we show that we can retain state-of-the-art machine learning performances on surfaces that are cheap to compute, effectively breaking the bottleneck for surface-based protein representation learning (Section 4).

## 2 MATHEMATICAL BACKGROUND

A $k$-simplex is defined as the convex hull of $k + 1$ affinely independent points. In $\mathbb{R}^3$, a 0-simplex is a point, a 1-simplex a segment, a 2-simplex a triangle, and a 3-simplex a tetrahedron. The face of a simplex is a simplex of smaller dimension defined by a strict subset of its vertices, and one defines a coface similarly. A simplicial complex is a collection of simplices closed under taking faces, such that the intersection of any two simplices is either empty or a face of each.

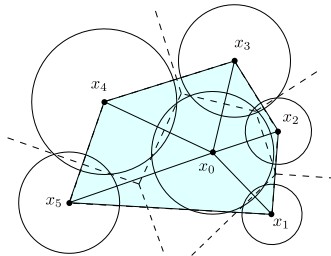 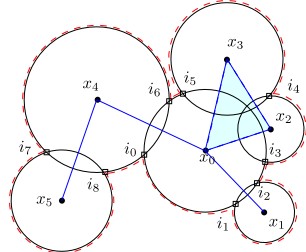

Figure 2: $\alpha$-complex and the boundary of a union of balls: 2D illustration. (Left) Regular triangulation of six balls (vertices, edges, cyan triangles), and the corresponding power diagram. (Right) Corresponding complex $\mathcal{K}_0$ (vertices, blue edges, cyan triangle), and boundary of the union of balls (red circle arcs). The ball centered at $x_0$ contributes three circle arcs–vertex $x_0$ is a regular 0-simplex in $\mathcal{K}_0$. The singular 1-simplex $(x_4, x_5)$ contributes the two intersection points $i_7$ and $i_8$ on the boundary.

Recall that the power distance from a point $x$ to a sphere $B_i(x_i, r_i)$ is defined by $\pi(x, B_i) = \|x - x_i\|^2 - r_i^2$. The *power diagram* of the balls generalizes the Euclidean Voronoi diagram by ascribing to the ball $B_i$ the set of points $V_i = \{x \in \mathbb{R}^3 \mid \forall j \neq i, \pi(x, B_i) \leq \pi(x, B_j)\}$. Intersections between these regions can in turn be used to define the so-called *regular triangulation*, a simplicial complex generalizing the usual Delaunay triangulation for balls. Indeed, two, three and four incident power regions define an edge, a triangle and a tetrahedron, respectively, in the regular triangulation. Remarkably, both diagrams are invariant upon considering the set of enlarged balls $\mathcal{S}_\alpha = \{B_i(x_i, \sqrt{r_i^2 + \alpha})\}_{i=1,\ldots,n}$ for $\alpha \geq 0$. This invariance enables defining the *restriction of a ball* as $R_{i,\alpha} = V_i \cap B_{i,\alpha}$. The $\alpha$-complex, denoted $\mathcal{K}_\alpha$, is the simplicial complex defined by:

$$\text{Simplex } \Delta \in \mathcal{K}_\alpha \iff \bigcap_{j \text{ vertex of } \Delta}^{k} R_{j,\alpha} \neq \emptyset. \tag{1}$$

Note that growing $\alpha$ results in larger restrictions, hence more simplices. For $\alpha$ large enough, $\mathcal{K}_\alpha$ therefore matches the regular triangulation.

## 3 MOLECULAR SURFACES

### 3.1 MOLECULAR SURFACES AND THEIR COMPUTATION

Several molecular surfaces have been defined to model a collection of atoms represented by balls/spheres of different radii (Connolly, 1983; 1985). The most natural one is plainly the outer boundary of the union of balls (Gavezzotti, 2007). This boundary surface consists of spherical caps contributed by the atoms, circle arcs contributed by intersecting spheres, and points found at the intersection of three spheres. This set of primitives is termed the *combinatorial structure* of the boundary of the union. This outer boundary defines the van der Waals surface when using the van der Waals radii, which are obtained from half the pairwise distance between atoms in organic molecules (Gavezzotti, 2007). For biomolecules whose interactions typically occur on *surface* patches *seen* by water molecules (Lee & Richards, 1971), the Solvent Accessible Surface (SAS) is defined as the boundary of the atomic balls enlarged by the radius of a water probe, taken as $r_w = 1.4$Å in general (Gerstein et al., 1995; Gerstein & Richards, 2001).

Another classical surface also obtained by rolling a water probe on the atoms is Connolly's surface, often called the molecular surface or solvent-excluded surface (SES) (Connolly, 1983). In addition to spherical caps contributed by the atoms, it also involves toroidal (resp. spherical, concave) patches contributed by the probe in contact with two (resp. three) atoms. This surface has the advantage of being smooth: a tangent plane exists everywhere.

Remarkably, the aforementioned molecular surfaces are encoded in selected simplices of $\mathcal{K}_\alpha$ (Akkiraju & Edelsbrunner, 1996). For example, the SAS surface reads directly from the so-called singular and regular simplices of $\mathcal{K}_0$ for the set of balls $\mathcal{S} = \{B_i(x_i, r_i + r_w)\}_{i=1,\ldots,n}$. The complexity to compute the $\alpha$-complexes is $O(n \log n + k)$, with k the number of simplices, empirically equal to $O(n \log n)$ for proteins (Cazals et al., 2011). An efficient and robust computation of $\alpha$-complexes is provided by the Computational Geometry Algorithm Library (cga)–see Appendix B.

Another avenue to compute molecular surfaces leverages atom properties. Indeed, atoms have comparable radii and their distances are lower bounded due to repulsion forces. For such a set, the number of edges and faces of the molecular surface is linear in the number of spheres and can be found in $O(n \log n)$ (Halperin & Overmars, 1994). This property is exploited by the `MSMS` program (Sanner et al., 1995) to compute the solvent-excluded surface.

Finally, triangulated molecular surfaces are often used for visualization purposes. Such meshes are delivered e.g. by `MSMS` (Sanner et al., 1995) and widely used in tools like PyMOL (DeLano et al., 2002). In a similar vein, the EDTSurf program projects the level set specification of the surface targeted (SAS, SES) onto a regular grid using the euclidean distance transform (Xu & Zhang, 2009). NanoShaper (Decherchi & Rocchia, 2013) does a similar projection using ray casting onto surfaces derived from the $\alpha$-complex computations. Both methods then extract triangular meshes from their respective grids using the marching cube algorithm.

We summarize the previous discussion by noting that there are *three* geometric structures associated with a molecular surface. The first one is the $\alpha$-complex which determines the combinatorial structure of the boundary of the union of atomic balls. This structure determines the molecular surface but also atomic packing (Gerstein & Richards, 2001) and interfaces (Janin et al., 2008; Loriot & Cazals, 2010) properties. The second is the combinatorial structure of the molecular surface itself, namely spherical and/or toroidal patches, connected by circle, arcs and points. These patches have been used to encode the geometry and topology of binding patches using so-called shelling orders, in a way amenable to efficient comparisons (Malod-Dognin et al., 2012; Cazals & Malod-Dognin, 2011). The last one is a triangle mesh approximating the molecular surface, as delivered e.g. by `MSMS` (Sanner et al., 1995). Such meshes have been used recently for learning purposes, *e.g.* in `MaSIF` (Gainza et al., 2020).

For learning applications, point clouds with approximate normals have been proposed as a surface proxy (Sverrisson et al., 2021), as well as coarsened versions of the triangular meshes returned by `MSMS`, (Mallet et al., 2024). Those modifications were introduced to limit the computational burden of a direct usage of the output of `MSMS`.

In encoding molecular properties, an interesting question is therefore to determine which modeling level is best, both in terms of accuracy and running time. Indeed, once the combinatorial structure has been computed, producing a mesh is an extra step. Moreover, the combinatorial structure being directly associated to the atoms themselves, the semantic information is accessible more readily.

*In the following, we focus on singular and regular triangles encoding the Solvent Accessible Surface and directly consider them as a mesh.*

## 3.2 Empirical comparisons

We start by comparing the different approaches used to build molecular surfaces. To do so, we use the dataset Masif-Ligand introduced by Gainza et al. (2020). This dataset contains 1438 proteins bound to one out of seven cofactors with sizes between 58 and 3188 residues.

We compare triangular meshes produced by `MSMS`, with and without coarsening, meshes formed directly by the regular simplices of the $\alpha$-complex for different values of alpha and the point clouds and normals proposed by dMaSIF(Sverrisson et al., 2021). We visualize them in Supplementary Figure 4. We report the time needed to compute their outputs as well as the number of vertices produced in Table 1, as well as the distribution of several properties in Supplementary Section D.

| Method | MSMS | | $\alpha$-complex | | dMaSIF |
| | Default | Coarsened | $\alpha = 0$ | $\alpha = 5$ | |
| --- | --- | --- | --- | --- | --- |
| Time (ms) | $961.0 \pm 831.0$ | $1356.0 \pm 826.5$ | $47.0 \pm 31.4$ | $45.0 \pm 26.5$ | $59 \pm 15$ |
| Size | $35994 \pm 24513$ | $3593 \pm 2441$ | $2456 \pm 1740$ | $1684 \pm 1129$ | $16254 \pm 11443$ |

Table 1: Computational performance and output complexity across surface construction methods. Timing values (ms) represent the mean and standard deviation for protein parsing and mesh generation and size represent the mean and standard deviation of the number of vertices/points

As seen, the $\alpha$-complex computation represents a significant 29x speed up in comparison to `MSMS` Coarsened while maintaining a lower vertex count. It even beats the creation of dMaSIF point clouds and normals. This demonstrates that $\alpha$-complex-based meshes offer a superior balance between reduced output complexity and rapid surface generation. Despite being coarser, those meshes also include less problematic acute triangles, as illustrated in Figure 5d.

## 4 LEARNING ON SURFACES

Learning on surfaces has emerged as a subfield of geometric deep learning with computer vision applications (Bronstein et al., 2017; Bruna et al., 2013; Masci et al., 2015; Sharp et al., 2022). Masif (Gainza et al., 2020) pioneered the application of such methods to molecules, extracting patches from `MSMS` triangulations. It was followed by several others directly using the whole triangulation along with coarsening (Mylonas et al., 2021; Riahi et al., 2023; Wang et al., 2023). Combining surface encoders with other modalities, such as geometric graphs, results in powerful embeddings (Mallet et al., 2024; Zhang et al., 2024; Qian et al., 2025; Zhou et al., 2025; Li et al., 2025). Atom-Surf (Mallet et al., 2024) couples a protein graph encoder Wang et al. (2022) and a surface mesh encoder Sharp et al. (2022) through message-passing neural networks to enable interaction between both modalities.

All the aforementioned rely on triangulations of the surface. However, once the combinatorial structure has been computed, producing a triangle mesh is an extra costly step resulting in large meshes. We believe the numerous small triangles present on the surface are an unnecessary proxy to the semantic information of the surface, incurring avoidable overheads. We hence propose a learning pipeline that replaces the meshes based on MSMS used in AtomSurf, by meshes directly extracted from the $\alpha$-complex (Figure 2), and name the resulting method AlphaSurf.

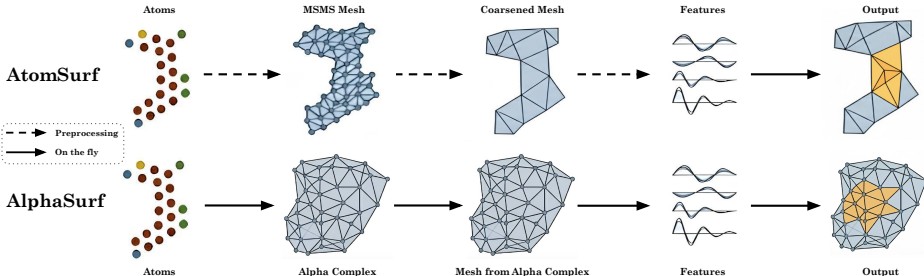

Figure 3: AlphaSurf operates entirely on-the-fly using $\alpha$-complexes instead of `MSMS` to get its mesh.

We evaluate our integrated pipeline on the aforementioned Masif Ligand dataset, aiming to classify binding pockets according to the cofactor they bind to (7 classes). We follow the exact architecture and learning setup as Mallet et al. (2024). We define the processing time for each protein as the time used for a forward and backward pass, computed for a batch of 8 proteins, on a V100 GPU. We define the preprocessing time as the time necessary to obtain the different surfaces. We report our timing and performance results in Table 2.

| Method | AtomSurf | AlphaSurf $\alpha = 0$ | AlphaSurf $\alpha = 5$ | dMaSIF |
|---|---|---|---|---|
| PDB parsing (ms) | $132 \pm 113$ | $132 \pm 113$ | $132 \pm 113$ | $132 \pm 113$ |
| Mesh generation (ms) | $1356 \pm 826$ | $47 \pm 31$ | $45 \pm 26$ | $59 \pm 15^{\dagger}$ |
| Featurization (ms) | $343 \pm 209$ | $273 \pm 147$ | $187 \pm 72$ | $7 \pm 1^{\dagger}$ |
| Preprocessing Time (ms) | $1831 \pm 867$ | $452 \pm 193$ | $364 \pm 140$ | $198 \pm 114^{\dagger}$ |
| Preprocessing Time (8 workers) (ms) | $540$ | $\underline{96}$ | $\mathbf{60}$ | $116^{\dagger}$ |
| Processing time (ms) | $800 \pm 120^{\dagger}$ | $720 \pm 110^{\dagger}$ | $630 \pm 110^{\dagger}$ | $8550 \pm 4250^{\dagger}$ |
| Balanced Accuracy (Top 1) | $\mathbf{86.2}$ | $86.1$ | $82.2$ | $80.0$ |
| Balanced Accuracy (Mean) | $\underline{83.0} \pm 3.0$ | $\mathbf{84.1} \pm 2.6$ | $81.7 \pm 0.5$ | $78.3 \pm 1.5$ |

Table 2: Framework comparison of efficiency, speed, and balanced accuracy. The mean/standard deviation is computed across three different seeds. $^{\dagger}$Computed on GPU; **Bold**: Best; Underline: Second best.

As can be seen, our method retains comparable balanced accuracy (average of recall on each class) with the `MSMS` triangulation and significantly outperforms proxies (86% vs 80% balanced accuracies). Conversely, using coarser meshes through larger alphas degrades performance. Importantly, AlphaSurf is substantially faster than AtomSurf (preprocessing time cut by a factor 5.3). Please note that model computations occur on the GPU, asynchronously from the data processing. Taking into account data processing parallelism on a standard GPU machine with 8 CPUs, our method reaches a data throughput on par with model computations.

## 5  DISCUSSION

Going back to the foundations of molecular surface computations, we notice that current surface-based learning methods could use coarser and faster surfaces. We implement this approach in AlphaSurf, retaining performance while drastically cutting preprocessing times. Our work allows on-the-fly surface computations, facilitating adoption of surface-based protein representation learning and opening the door to surface data augmentation.

Our work also includes several limitations. Further validation on larger datasets are needed to confirm our initial results. Moreover, using smaller models and fewer eigenvectors would result in an even lighter approach, cutting the runtime by another order of magnitude. Investigating the performance of such an approach represents an interesting follow-up.

### MEANINGFULNESS STATEMENT

Surface-based representations being increasingly used and resulting consistently into performance boosts, we believe our contribution will significantly facilitate their adoption and eventually contribute to learning meaningful representations of life.

### ACKNOWLEDGMENTS

V.M. is supported by a Junior Springboard Prairie program, funded by the ANR project ANR-23-IACL-0008. This work was performed using HPC resources from GENCI–IDRIS (Grant AD010315435R1). F.C. is supported by the 3IA Côte d'Azur 2030 (grant ANR-23-IACL-0001).

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

APPENDIX

## A CODE

The code is available here https://github.com/VictorGtn/AlphaSurf

## B FURTHER MATHEMATICAL EXPLANATIONS ON SURFACE COMPUTATIONS

**Relation between the $\mathcal{K}_\alpha$ and the boundary of a union of balls** The theory of weighted $\alpha$-complexes is non trivial, see (Edelsbrunner, 1992, Table 5.1) and (Akkiraju & Edelsbrunner, 1996). Assume we want to compute the boundary $\partial \cup_i B_i$ of a collection of balls $\{B_i\}$. As mentioned in the main text, we use $\mathcal{S} = \{B_i(x_i, r_i + r_w)\}$ to define the SAS. In the following, we just provide the link between the so-called *singular* and *regular* simplices of $\mathcal{K}_0$ for these balls, and $\partial \cup_i B_i$.

In short, singular and regular simplices make the following contributions to $\partial \cup_i B_i$:

A *singular* simplex in $\mathcal{K}_\alpha$ is a simplex which does not have any coface. That is, this simplex is not the face of a higher dimensional simplex. Cases of interest contributing to $\partial \cup_i B_i$ are:

- singular vertex: contributes a full sphere. (Does not happen in proteins since all protein atoms have covalent bonds.)

- singular edge: contributes the full circle defined by two intersecting spheres. (Typically happens to atoms at the *tip* of side-chains.)

- singular triangle: contributes the two intersection points defined by its three spheres. (Happens in *thin* regions of the proteins, when atoms form a layer.)

A *regular* simplex in $\mathcal{K}_\alpha$ is a simplex which is not singular and satisfies either condition: it belongs to the convex hull (the boundary of the Delaunay/regular triangulation), or it belongs to the interior and does not have all its cofaces in $\mathcal{K}_\alpha$. Cases of interest contributing to $\partial \cup_i B_i$ are:

- regular vertex: contributes one or several spherical caps.

- regular edge: contributes one or several circle arcs.

- regular triangle: contributes one of the two intersection points defined by its three spheres.

**Computing the $\alpha$-complex.** It should also be noted that the computation of the (regular) Delaunay triangulation can be unstable, as rounding errors in (nearly) degenerate configurations confuse the logic of the algorithms (Kettner et al., 2008). For these reason, the effective calculations of $\alpha$-complexes only became possible with the engineering on number types and geometric predicates, as performed within the Computational Geometry Algorithms Library (CGAL) (cga). In practice, we use the Fixed_alpha_shape_3 class from CGAL.

## C VISUALIZATION OF THE $\alpha$-COMPLEX OF A MOLECULE

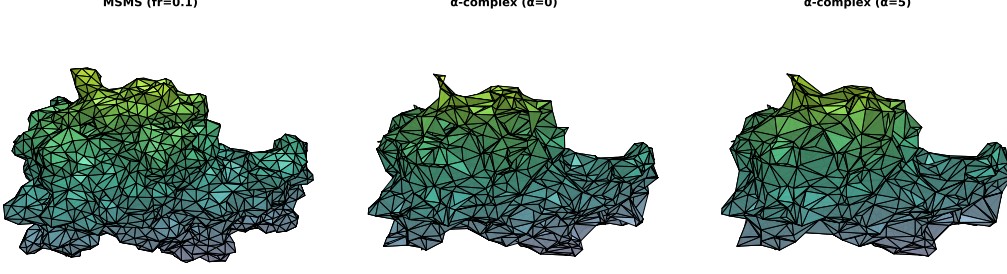

Figure 4: Geometric representations of the 1A27_AB protein complex: a) MSMS coarsened mesh, b) $\alpha$-complex with $\alpha = 0$, and c) alpha complex with $\alpha = 5$.

# D DISTRIBUTION OF THE MESHES

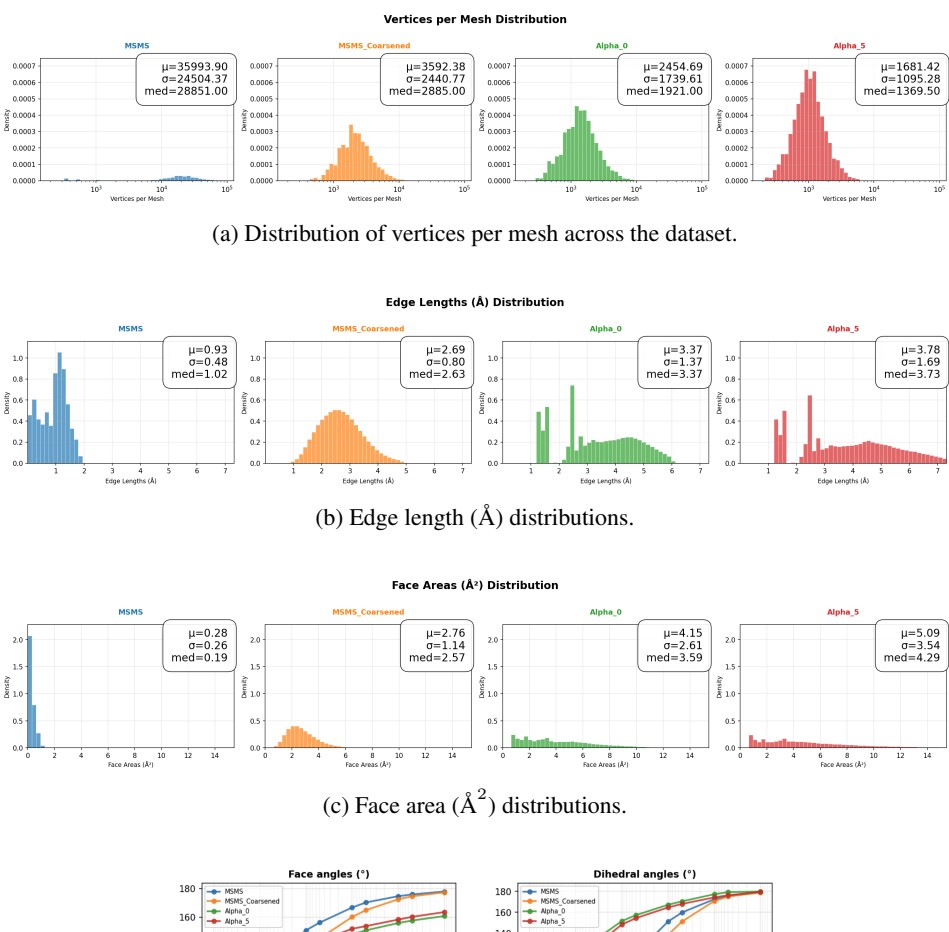

(a) Distribution of vertices per mesh across the dataset.

(b) Edge length (Å) distributions.

(c) Face area (Å$^2$) distributions.

(d) Conditional mean of the top $N\%$ for face angles.

Figure 5: Analysis of mesh geometry: vertex counts, edge/face metrics, and angular distributions.

**Distribution of Vertex Numbers:** The distribution of MSMS is wide, with large meshes and significant fluctuation. As seen, as $\alpha$ increases, the resulting mesh is becoming coarser. All in all, this suggests that $\alpha$-shapes are less detailed than MSMS.

**Distribution of Edge Lengths:** MSMS Coarsened provides a smoother edge length distribution than MSMS. For $\alpha$=0 and $\alpha$=5, the distribution is bimodal with a first mode in $[1.33; 2]$ corresponding to atoms located at the surface having covalent bonds, and a second smoother mode whose length depends on the $\alpha$.

**Distribution of Face Areas:** MSMS Coarsened provides a smoother face area distribution than MSMS. Moreover, MSMS has a clear spike around 0Å$^2$ which indicates a high frequency of sliver triangles (triangles with near-zero area), making it even more predatory to learn even disregarding the computational issues. Both $\alpha$=0 and $\alpha$=5 have larger distributions than MSMS coarsened, shifting the density toward higher values as the $\alpha$ parameter increases. This shift reflects the geometric relaxation inherent in the $\alpha$-shape construction

**Conditional Distributions**    As we look at the "worst" $N\%$ face angles (the top right of the graph), the `MSMS` and `MSMS` Coarsened face angles climb much faster toward 180°. An angle of 180° means the triangle has collapsed into a line. This is a predatory behavior regarding learning on meshes, which does not happen on the meshes derived from the $\alpha$-complex. The right plot confirms the visual impression that $\alpha$-complexes provide rougher mesh than `MSMS`, as seen in Figure 4. This reflects the fundamental difference in surface representation: $\alpha$-complexes prioritize topological correctness, which can produce sharp angular transitions, while `MSMS` optimizes for surface smoothness.

