# OpenReview forum: "ALPHASURF: ON-THE-FLY SURFACE COMPUTATIONS FOR PROTEIN REPRESENTATION LEARNING"
_ICLR.cc/2026/Workshop/LMRL — ICLR 2026 Workshop LMRL Poster_

### Official Review · Reviewer_okdJ · 2026-02-13
**Engineering utility for protein surface representation learning methods**

**Rating:** 6
**Confidence:** 4

**Review:**

**Summary**:
This paper introduces AlphaSurf, a method that replaces slow, traditional molecular surface generation with fast α-complex meshes to eliminate the computational bottleneck in protein representation learning. By leveraging the mathematical properties of weighted α-shapes, the authors demonstrate a large speedup in mesh generation, allowing protein surfaces to be computed "on-the-fly" during model training. When integrated into the state-of-the-art AtomSurf architecture, AlphaSurf maintains high classification accuracy on a single binding pocket task while significantly reducing preprocessing time and storage needs. The work shows that coarser, topologically robust meshes provide an efficient and numerically stable representation for geometric deep learning. TLDR: The authors turn to α-complexes for more efficient mesh development for already developed DiffusionNet architectures.

**Strengths**
- Practical Utility: The removal of the "MSMS bottleneck" is a quality-of-life improvement. Pre-computing meshes is a barrier to entry that this paper lowers.
- Efficiency: Matching the speed of "point-cloud" methods (like dMaSIF) while retaining the performance of "mesh" methods (like AtomSurf) is a strong empirical result.

**Weaknesses**
- The paper tests $\alpha=0$ and $\alpha=5$. $\alpha$ is a continuous parameter that defines the "tightness" of the mesh. This is missing an ablation study explaining why $\alpha=0$ the sweet spot?
- The only experiment is run on pocket classification (MaSIF-Ligand) is a relatively "easy" task for surface models because pockets have distinct concave signatures. This is a task that likely will be less impacted by changing the coarseness of the surface mesh, so having other tasks would be necessary to show broad use. Using the PINDER benchmarking tasks from the original AlphaSurf paper would strengthen this work greatly.
-  No comparisons against other modern, fast meshing techniques like EDTSurf or NanoShaper, which are significantly faster than MSMS but smoother than $\alpha$-complexes. This makes the "AlphaSurf is the only fast option" argument the authors make feel slightly cherry-picked. If you can still have just as fast meshing but smoother is that actually better or worse for these benchmarking tasks? This is not clear.

---

### Official Review · Reviewer_csxz · 2026-02-25
**Valuable Contribution Requiring Clarification**

**Rating:** 8
**Confidence:** 3

**Review:**

This paper proposes to use the simplices from $\alpha$-complexes directly as molecular surfaces, which can be used as features in deep learning models. The authors test their idea on a classification task and demonstrate that their surfaces require less time to compute while maintaining high performance.

Pros:
One positive point is the simplicity of the proposed method, which allows skipping common steps (triangulation, coarsening) that are a computational burden. Also, this paper has the potential for high impact in the field by greatly reducing computation resources without sacrificing quality of features.

Cons:
The paper's organization should be improved to increase clarity. As it is, Section 2 on mathematical background currently lacks motivation, as the task of molecular surface computation is not yet well defined. For the same reason, the description of competing approaches should be improved: DmaSIF should be defined before line 136 (add an explicit mention to DmaSIF at line 120, and clarify whether DmaSIF approximates the SAS or another surface). MSMS should be defined properly to avoid confusion (line 104 introduces MSMS in the context of solvent excluded surfaces, and it is not mentioned that it can also produce approximations to the SAS, and how). More benchmarking on larger and different datasets is needed to evaluate the method more robustly and to convice readers of the usefulness of the method. More discussion is needed, particularly about the impact of smoothness of surfaces on downstream tasks, and on the impact of alpha on the resulting surface.

I can't judge the originality of the work because I am not familiar with the field, but according to the claims of the paper, the proposed method is new.

Minor comments:
- the definition of simplicial complex is missing the closure under taking faces.
- Figure 1 is not referenced in the text, and lacks a description of which dataset/task it represents.
- the symbols and scripts used in the tables (boldface, underlined, cross superscript) are not defined.
- In Figure 2, showing a diagram of $K_\alpha$ with a greater $\alpha$ would help explain the method better.
- I didn't understand what is meant by the word "contribute" in line 067, since the intersection points are not part of $K_\alpha$.
- PDB is undefined at line 146.
- the claim at line 153 is lacking evidence to support it.
- I would recomment adding a total time row in Table 2, and additionally reporting regular (unbalanced) accuracy in tables (possibly in supplemental)
- At line 181, the setup/architecture should be described in a few words.

---

### Official Review · Reviewer_fiaP · 2026-02-25
**A highly interesting paper that addresses computational complexity of surface-based protein representation learning**

**Rating:** 9
**Confidence:** 4

**Review:**

The paper addresses an often-ignored but important aspect of contemporary machine learning: Computational efficiency. The authors propose replacing Sanner's MSMS with an on-the-fly approach to compute molecular surfaces, which is considerably faster by simply limiting itself to computing $\alpha$-complexes and discarding mesh generation.

$\textbf{Strenghts}$

The presented method is an elegant way to significantly increase the computational efficiency of protein representation learning. I strongly believe that this alone warrants inclusion in a workshop of a field where computational efficiency is too often ignored.

The analysis of the mesh distributions in the appendix is an important contribution that also uncovers limitations of the MSMS approach, and I consider it the main result of the work. It should feature in the main.

$\textbf{Weaknesses}$

A weakness that may be identified by some is the lack of comparison against non-surface-based representations. While this is technically true, I believe that this does not detract from this paper as it addresses the (computational) performance questions of this subset of methods.

What should have been addressed by the authors are the potential failure modes of the approach. For example, could the more coarse-grained approach fail to identify the binding pockets of specific proteins with relatively complex geometries?

Overall, I believe this to be an excellent workshop paper that addresses an understudied problem: Computational efficiency in (biology) applied machine learning.

---

### Meta-Review · Area_Chair_wUrw · 2026-02-28

**Recommendation:** Accept (Poster)
**Confidence:** 4

**Metareview:**

Given the tiny paper space limitations this is a very nice contribution, and happy to have it discussed at the workshop.

---

### Decision · Program_Chairs · 2026-03-02

**Decision:**

Accept (Poster)

**Comment:**

Please see the meta-review.